# Pharmacological inhibition of TBK1/IKKε blunts immunopathology in a murine model of SARS-CoV-2 infection

Tomalika R. Ullah [1,2,17], Matt D. Johansen [3,17], Katherine R. Balka [4], Rebecca L. Ambrose[1,2], Linden J. Gearing [1,2], James Roest[5], Julian P. Vivian[5,6], Sunil Sapkota[1,2], W. Samantha N. Jayasekara[1,2], Daniel S. Wenholz[7,8], Vina R. Aldilla[8], Jun Zeng[9], Stefan Miemczyk [3], Duc H. Nguyen[3], Nicole G. Hansbro[3], Rajan Venkatraman[4], Jung Hee Kang[4], Ee Shan Pang[4], Belinda J. Thomas[1,2,10], Arwaf S. Alharbi [1,2,11], Refaya Rezwan[1,2], Meredith O'Keeffe [4], William A. Donald [8], Julia I. Ellyard [12,13], Wilson Wong[1,2,14], Naresh Kumar[8], Benjamin T. Kile[4,15], Carola G. Vinuesa [12,13,16], Graham E. Kelly[7], Olivier F. Laczka [7], Philip M. Hansbro [3,18], Dominic De Nardo [4,18] & Michael P. Gantier [1,2,18] ✉

TANK-binding kinase 1 (TBK1) is a key signalling component in the production of type-I interferons, which have essential antiviral activities, including against SARS-CoV-2. TBK1, and its homologue IκB kinase-ε (IKKε), can also induce pro-inflammatory responses that contribute to pathogen clearance. While initially protective, sustained engagement of type-I interferons is associated with damaging hyper-inflammation found in severe COVID-19 patients. The contribution of TBK1/IKKε signalling to these responses is unknown. Here we find that the small molecule idronoxil inhibits TBK1/IKKε signalling through destabilisation of TBK1/IKKε protein complexes. Treatment with idronoxil, or the small molecule inhibitor MRT67307, suppresses TBK1/IKKε signalling and attenuates cellular and molecular lung inflammation in SARS-CoV-2-challenged mice. Our findings additionally demonstrate that engagement of STING is not the major driver of these inflammatory responses and establish a critical role for TBK1/IKKε signalling in SARS-CoV-2 hyper-inflammation.

TBK1 is a point of convergence for several key pattern recognition receptor pathways, including cGAS-STING, Toll-like receptors (TLR)3 and 4 (via TRIF), and RIG-I-MAVS. Following its activation, TBK1 phosphorylates STING/TRIF/MAVS to recruit and activate IRF3 resulting in the induction of type I interferons (IFNs) (e.g., IFN-β)[1]. While normally protective against viral infection, TBK1-driven IRF3 transcription can also fuel viral-driven hyperinflammation and auto-immune diseases arising from aberrant engagement of STING, TRIF, or MAVS[2,3]. Several studies suggest that pharmacological inhibition of

TBK1 kinase activity could alleviate aberrant inflammation stemming from these pathways, but whether this could be applicable in the context of viral-driven hyper-inflammation such as that emanating from SARS-CoV-2 infection is not defined[4,5].

TBK1 activation is entirely dependent on phosphorylation of serine 172 (S172)[6], induced upon interaction with STING/TRIF/MAVS[1]. To-date, small molecule inhibitors of TBK1 and its analogue IKKε (e.g. amlexanox, BX795, MRT67307 or WEHI-122), have specifically targeted their kinase activities[5,7]. While potently blocking IRF3 activation, this

does not prevent TBK1 S172 phosphorylation[8], nor block the STING-mediated nuclear factor (NF)-κB transcriptional programme that TBK1 controls redundantly with IKKε[9].

Here we report an alternative inhibitor of TBK1/IKKε signalling that disrupts the formation of TBK1/IKKε signalling complexes, reducing phosphorylation of S172 and resulting in dual inhibition of both IRF3 and NF-κB transcriptional programmes. Importantly, we demonstrate that both non-classical and classical inhibition of TBK1/IKKε are more protective than direct STING inhibition against severe SARS-CoV-2-driven immunopathology in vivo, showing unequivocally the therapeutic potential of targeting TBK1/IKKε to limit SARS-CoV-2 hyper-inflammation.

## Results
### Idronoxil inhibits STING, MAVS and TRIF driven responses
Flavonoids are a highly diverse class of plant metabolites with a wide range of health benefits, including widespread anti-inflammatory effects[10,11]. Recent evidence suggests that a select subset of flavonoid and related compounds (e.g., epigallocatechin gallate, resveratrol, and genistein), can inhibit inflammation through the cGAS-STING-TBK1 pathway[12–14]. Here, we initially observed that idronoxil (IDX), a synthetic flavonoid closely related to daidzein, stood out among other flavonoid compounds as a potent inhibitor of STING-induced IRF3 and NF-κB transcriptional programmes upon cGAS overexpression (in HEK-STING cells), or STING-induced signalling in both human and murine cell lines (mouse L929 cells, mouse immortalised bone marrow-derived macrophages [iBMDMs], THP-1 cells, and hTERT-immortalised human foreskin fibroblasts[15]; Fig. 1a–c and Supplementary Fig. 1a–f). Accordingly, IDX significantly inhibited IRF3-driven IP-10 and IFN-β production, while also dampening NF-κB-driven IL-6 and TNF production upon STING activation (Fig. 1a, c and Supplementary Fig. 1a–c, e, f). In addition, IDX broadly decreased an IFN-stimulated gene (ISG) signature upon constitutive engagement of the cGAS-STING pathway in primary BMDMs derived from $Trex1^{Q98X}$ mutant mice (Fig. 1d)[16]. Consistent with these findings, phosphorylation of STING, TBK1, IKKε, and consequently downstream IRF3 and p65 was reduced following STING stimulation in iBMDMs and hTERT fibroblasts in the presence of IDX (Fig. 1e and Supplementary Fig. 1g). These data suggest that IDX has a direct impact at the level of STING, or on TBK1 and IKKε[9].

To examine the potential specificity of IDX for STING signalling, we next tested its impact on the TRIF-IRF3 branch driven by TLR4 and TLR3 agonists, using lipopolysaccharide (LPS) and poly(I:C), respectively. As shown in iBMDMs and hTERT fibroblasts, IDX significantly decreased TLR4- and TLR3-driven IP-10 production, indicating inhibition of the TRIF-TBK1-IRF3 axis (Supplementary Fig. 1h–j). The inhibitory activity of IDX on TLR4 signalling appears to operate at the level of TRIF/TBK1/IKKε, as revealed by the reduction in TBK1 or IKKε S172 phosphorylation in WT and TBK1-deficient iBMDMs, respectively (Fig. 1f). Similarly, IDX inhibited MAVS-driven IRF3 activity upon stimulation of HEK cells with a selective RIG-I agonist, or upon transfection of THP-1 cells with poly(I:C), assessed with IFN-β-Luciferase reporter expression and IP-10 production, respectively (Fig. 1g and Supplementary Fig. 1k). Critically, dose-dependent inhibitory effects of IDX on type-I IFN production following STING stimulation were not observed in TLR7-driven IFN-α production (a TBK1-independent pathway[17]) from splenic plasmacytoid dendritic cells (Supplementary Fig. 1l). In addition, comparable p65 phosphorylation levels in wild-type (WT) and TBK1/IKKε-double deficient cells indicated that IDX did not impact TLR4-driven NF-κB signalling, which is known to be independent of TBK1/IKKε[9] (Supplementary Fig. 1m). Finally, both IDX and the commercial TBK1/IKKε small molecule kinase inhibitor, MRT67307 (MRT hereafter) significantly decreased TLR3-, cGAS-, and RIG-I/MDA5-driven *IFNB1* and ISG expression in human primary

bronchial epithelial cells (PBECs), noting that IDX was generally more potent at the doses used (Fig. 1h).

Collectively, these results demonstrate that IDX has significant inhibitory activity on STING-, TRIF- and MAVS-driven signalling pathways either by directly targeting these adaptors, or via TBK1/IKKε.

### Idronoxil inhibits formation of TBK1 signalling complexes
Since TBK1/IKKε are common components in all the pathways impacted by IDX, we hypothesised they are the most likely molecular targets. Hence, we next assessed the effects of IDX on an IFN-β-Luciferase reporter driven by overexpression of TBK1 or IKKε (Fig. 2a). IDX strongly inhibited IFN-β-Luciferase driven by both kinases, with a noticeably greater inhibitory effect on IKKε signalling (IC50 of 0.49 μM for IKKε versus 0.96 μM for TBK1) (Fig. 2a). Aligning with this observation, surface plasmon resonance (SPR) analyses experimentally supported a weak interaction ($K_D = 39.16\,\mu M$) between IDX and recombinant TBK1 in vitro (Fig. 2b and Supplementary Fig. 2a). We note that the affinity of the interaction between IDX and TBK1 was relatively low in view of its potency in the micromolar range in cells. While this discrepancy could relate to conformational changes of TBK1 induced through interaction with other protein binding partners in cells, and which are lacking in our in vitro assays, these data confirmed that IDX can bind TBK1. Crucially however, IDX did not inhibit the kinase activity of TBK1 or IKKε, unlike MRT (Fig. 2c and Supplementary Fig. 2b)[8,9], implying that IDX has an alternative inhibitory mechanism of action on TBK1/IKKε function.

Interaction between the C-terminal tail of STING and TBK1 homodimers is essential for TBK1 kinase activation and its subsequent phosphorylation of STING and IRF3[18]. In silico modelling of IDX binding to a TBK1 dimer suggested that IDX binds at the groove formed between the two TBK1 monomers (Fig. 2d and Supplementary Fig. 2c). Such an interaction could potentially interfere with TBK1 binding to the C-terminal tail of STING[19]. This model of IDX-TBK1 interaction was concordant with structure-activity relationship for IDX (Supplementary Fig. 2d, e). Thus, both phenol groups of IDX (**1**) were essential for STING inhibition (as observed with **2**, **3**, **4** and **7**—Supplementary Fig. 2d, e). Computational modelling predicted the A-ring and B-ring phenols to undergo hydrogen bond interactions with Met1 and Gln581, respectively, interactions which would be disrupted in these four IDX analogues. Changing the 2*H*-chromene core to a coumarin core (in **5**) also resulted in reduced inhibitory activity, possibly due to a disruption of a carbon-hydrogen bond between the 2-position of IDX with Asn578. Addition of a phenol to the 8-position of the A-ring (**6**) similarly reduced the inhibitory activity, aligning with this group undergoing an intramolecular hydrogen bonding with the 7-OH group to impair interaction with Met1.

Critically, pull-down analyses of mCitrine-tagged murine STING confirmed experimentally that IDX reduced the interaction between mCitrine-STING and endogenous TBK1 and IKKε following DMXAA-stimulation in iBMDMs in a dose-dependent manner (Fig. 2e). In line with a different modality of action for IDX, classical TBK1 inhibition with MRT did not decrease the interaction between STING and TBK1, and instead increased phosphorylation of S172 in agreement with prior reports (Fig. 2f)[9]. Interestingly, both MRT and IDX reduced the STING-IKKε interaction, which we hypothesised could involve the inhibition of TBK1 function by both small molecules in these WT cells (Fig. 2f). Nonetheless, IDX directly inhibited phosphorylation of STING, IKKε, and downstream p65 in TBK1-deficient cells, where STING signalling entirely relies on IKKε[9] (Supplementary Fig. 2f). This is in contrast to published findings with WEHI-112, a TBK1/IKKε kinase inhibitor similar to MRT[5], which did not inhibit phosphorylation of STING, IKKε, or downstream p65 in TBK1-deficient cells[9]. This suggests that IDX can directly inhibit IKKε, independent of TBK1, through an alternative mechanism to that of classical kinase inhibitors. In agreement with these findings, IDX inhibited STING-driven IP-10 production in TBK1-

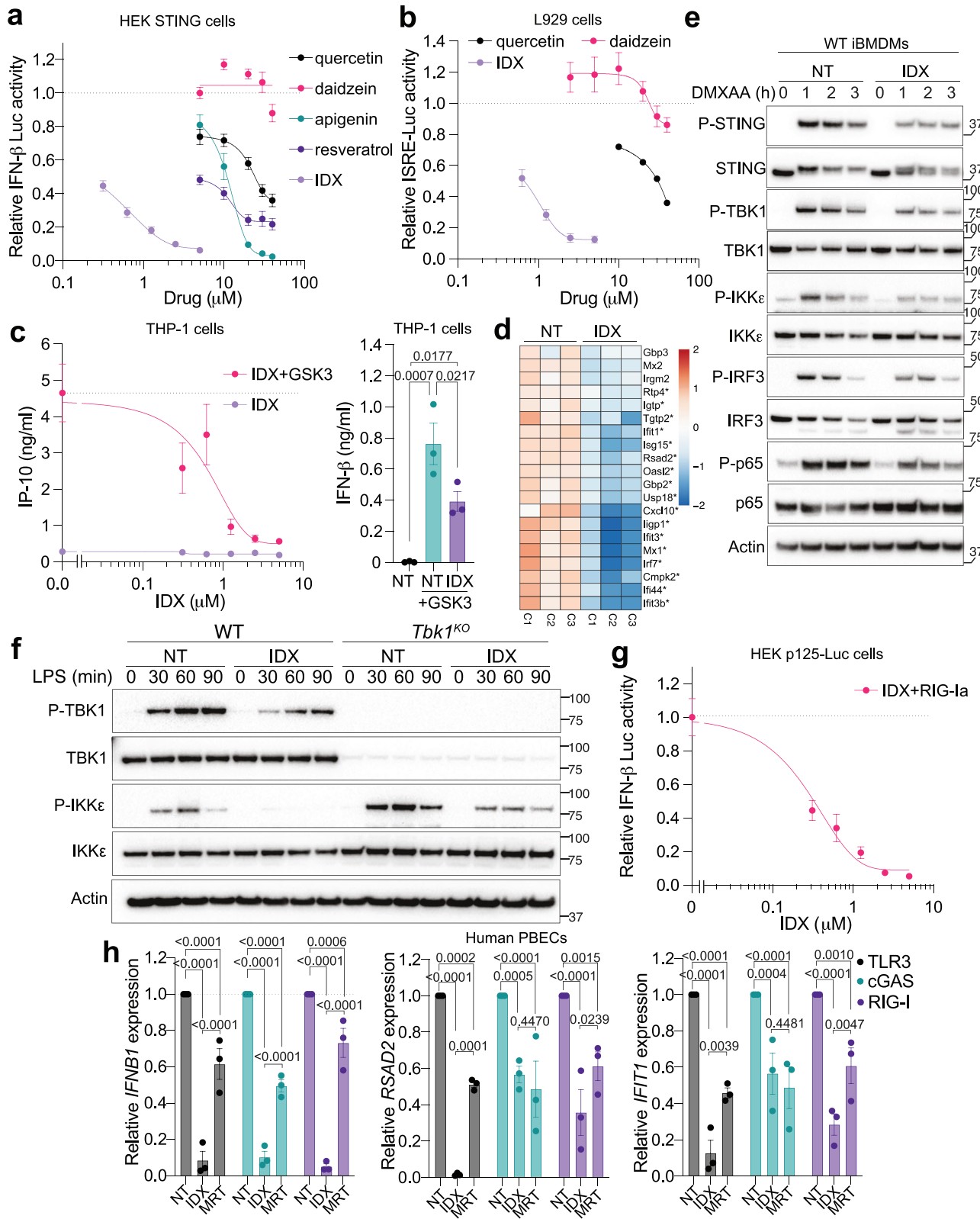

deficient cells, which was not observed with MRT (Supplementary Fig. 2g). We also observed the reciprocal results in IKKε-deficient cells, demonstrating that the inhibitory activity of IDX on TBK1 can occur independently of IKKε (Supplementary Fig. 2f). In agreement with our overexpression assays (Fig. 2a), IDX-dependent inhibition of STING-driven IP-10 production was more potent in TBK1-deficient cells (relying on IKKε), compared to WT or IKKε-deficient cells (relying on

TBK1 Fig. 2g). In addition, while IDX inhibited both IFN-β and NF-κB Luciferase reporters driven by the TLR3/TRIF axis in HEK-TLR3 cells, the inhibitory activity of MRT was restricted to the IRF3/IFN-β arm and did not decrease TLR3-driven NF-κB signalling (Fig. 2h), further high-lighting a differential mode of action.

Altogether, these results identify a non-classical inhibitory mechanism-of-action for IDX, showcasing an ability to disrupt TBK1/

**Fig. 1 | Idronoxil inhibits STING, MAVS and TRIF driven responses. a** Interferon [IFN]-β luciferase expression in HEK293T mSTING cells overexpressing cGAS treated O/N with the indicated compounds (*n* = 3 independent experiments). **b** IFN-stimulated response element [ISRE]-Luciferase expression in L929 cells after 7 h stimulation with the mouse STING agonist DMXAA at 20 μg/ml and the indicated compounds (*n* = 3 independent experiment for all except Quercetin *n* = 2 independent experiments). **c** IFN-β (right) and IP-10 (left) protein levels in THP-1 monocytes after 6 h stimulation with the human STING agonist GSK3 at 100 nM and treatment with indicated idronoxil (IDX) doses. Right: 2.5 μM IDX was used (*n* = 3 independent experiments). **d** RNA sequencing analysis of the top 20 murine ISGs[61] in RNA lysates from *Trex1*^Q98X bone marrow-derived macrophages [BMDMs] treated O/N with 1.25 μM IDX (heat map of log₂ expression relative to the average across all samples; significant genes are indicated with an asterisk) (*n* = 3 independent animals [C1–C3]). NT is non-treated. **e** Immunoblot of wild-type [WT] immortalised bone marrow-derived macrophages [iBMDMs] stimulated with 50 μg/ml DMXAA (STING) for the indicated times ±2.5 μM IDX (1 representative blot of 3 independent experiments shown). Molecular weight markers are shown in kDa. A doublet of STING bands appears upon STING activation. Phosphorylated proteins are indicated with [P]. **f** Immunoblot of wild-type and *Tbk1*^KO iBMDMs stimulated with 200 ng/ml LPS (TLR4) ± 2.5 μM IDX treatment for the indicated time (1 representative blot of 3 independent experiments shown). Phosphorylated proteins are indicated with [P]. Molecular weight markers are shown in kDa. **g** IFN-β luciferase expression in p125HEK cells treated with indicated dose of IDX and stimulated O/N with 50 ng/ml of 3p-hpRNA (RIG-I agonist [RIG-Ia]) (*n* = 3 independent experiments). **h** RT-qPCR analyses of *IFNB1/18S*, *RSAD2/18S* and *IFIT1/18S* in RNA lysates from primary human bronchial epithelial cells [PBECs] from 3 independent donors treated with 5 μM IDX or 200 nM MRT, and stimulated 3 h with 1 μg/ml transfected polyI:C (MDA5/RIG-I), 25 μg/ml naked polyI:C (TLR3) or transfected with 2.5 μg/ml ISD70 (cGAS) (*n* = 3 independent donors). **a**–**c**, **g**, **h** Data are mean ± s.e.m. **c** One-way or (**h**) two-way ANOVA with uncorrected Fisher's LSD (with single pooled variance) multiple comparisons are shown. **a**–**c**, **g** Non-linear regression analyses are shown. Exact *p* values are shown for all comparisons. Source data and detailed statistical analyses are provided as a Source Data file.

IKKε protein complexes, decrease S172 phosphorylation, and reduce downstream NF-κB- and IRF3-dependent inflammatory signalling following activation of STING, TLR3, TLR4, and RIG-I.

## Idronoxil protects against SARS-CoV-2-driven hyper-inflammation

While initially protective against SARS-CoV-2 infection in the upper airways, sustained/delayed IFN production in the lung has been proposed to fuel hyper-inflammation seen in patients with severe COVID-19[20,21]. In addition to contributing to the SARS-CoV-2-driven antiviral response in infected cells through MAVS-TBK1 signalling[22], recent evidence indicates that TBK1 signalling directly contributes to deleterious hyper-inflammation via engagement of the cGAS-STING pathway upon mitochondrial DNA sensing in damaged cells[23,24]. We found that a solubilised form of IDX injected intraperitoneally (i.p.) in mice, while mostly cleared from systemic circulation by 12 h, was retained in the lung in a dose-dependent manner (Supplementary Fig. 2h).

Therefore, to assess the therapeutic potential of TBK1/IKKε inhibition by IDX in reducing severe pulmonary hyper-inflammation, we used a lethal SARS-CoV-2 mouse model with transgenic expression of ACE2 (K18-hACE2 mice)[25]. The treatment regimen consisted of one daily i.p. injection of solubilised IDX starting from 3 days post-SARS-CoV-2 infection and preceding the onset of clinical signs, which continued until the experiment was concluded on day 6. Mice injected with IDX lost significantly less weight than those injected with the vehicle control (Fig. 3a). Critically, IDX administration significantly decreased total neutrophil and lymphocyte levels in the bronchoalveolar lavage fluid (BALF), although it did not reduce total leukocyte or macrophage levels (Fig. 3b). Consistent with these observations, lung histological analyses revealed a significant reduction in both the inflammatory score and airway collagen deposition in the IDX-treated SARS-CoV-2 group compared to the vehicle control group (Fig. 3c–f), directly supporting a protective effect of IDX against severe pulmonary inflammation and airway fibrosis. Accordingly, IDX treatment significantly limited the production of pro-inflammatory IL-6 and TNF in lung homogenates, without affecting viral titres (Fig. 3g and Supplementary Fig. 3a). Consistent with the observed decrease in airway neutrophils in the BALF, CXCL1 levels in lung homogenates and BALF were also significantly decreased following IDX treatment (Fig. 3g and Supplementary Fig. 3b). Lastly, profiling of IFN-regulated gene expression in the lung revealed a significant reduction in *Ifng*, *Irf7*, and *Oas3* expression in the IDX-treated SARS-CoV-2 group (Fig. 3h). Conversely, IDX treatment prevented a decrease in *Ifnar2* and *Ifngr1* expression, which are both essential for type-I and type-II IFN signalling, but did not significantly alter the expression of other antiviral genes (e.g., *Ifnb1*, *Oas1b*, *Oas2*, or *Mx2*) at 6 days post-infection (Fig. 3h and Supplementary Fig. 3c), noting the dynamic expression of these

genes in this model[26]. Finally, IDX showed no direct antiviral activity against SARS-CoV-2 using in vitro infection of Vero cells (Supplementary Table 1), confirming that the effects seen in vivo are related to its anti-inflammatory activity in host immune cells.

## TBK1 inhibition limits SARS-CoV-2-driven hyper-inflammation

Recent evidence has demonstrated that therapeutic administration of the STING palmitoylation inhibitor, H151, from 2 days post-SARS-CoV-2 infection, resulted in decreased weight-loss, lung inflammation, and antiviral gene expression, as well as increased survival of K18-hACE2 mice[24]. To define whether therapeutic targeting of TBK1 was superior to that of STING in the pathogenesis of severe SARS-CoV-2 infection, we next repeated a head-to-head comparison of the protective effect of IDX, MRT, and H151 in K18-hACE2 mice (noting that this experiment was terminated on day 5 post-infection due to several mice reaching the ethical endpoint across the different cohorts). After 5 days, H151 was the only treatment that significantly protected the animals against virus-induced weight loss (Fig. 4a). However, both IDX- and MRT-treated mice broadly showed a greater reduction in immune cell infiltrate in the BALF (Fig. 4b), which correlated with a significantly decreased inflammatory score (by histology) for both TBK1 inhibitors that was not seen with H-151 (Fig. 4c). All three inhibitors reduced alveolar thickness—indicative of decreased pulmonary inflammation and airway fibrosis—to normal levels (Fig. 4d). Accordingly, all three treatments significantly decreased the production of pro-inflammatory CXCL1 in lung homogenates, but only TBK1/IKKε inhibition significantly limited the production of IL-6 and MCP-1 at this time-point (Fig. 4e and Supplementary Fig. 4a). Similarly, induction of antiviral gene expression (e.g., *Oas1b*, *Oas2*, *Oas3*) in the lung was significantly reduced by both TBK1/IKKε inhibitors but not by STING inhibition (Fig. 4f). *Ifng* induction was also significantly lower with IDX and MRT compared to H151 (Supplementary Fig. 4b). It is also noteworthy that MRT was slightly more inhibitory than IDX on select ISGs, including *Irf7*, *Mx2*, and *Ifngr1* (Supplementary Fig. 4b). Importantly, induction of *Ifnb1* and *Ifnl3* levels in the lung was not impacted by any of the treatments (Supplementary Fig. 4b). Consequently, the viral titres in the lung (both BALF and homogenates) were not significantly impacted by either TBK1/IKKε or STING inhibition. Nevertheless, both IDX and MRT significantly increased the viral load in brain homogenates at day 5 (Supplementary Fig. 4c), suggesting that TBK1/IKKε systemic inhibition could increase viral encephalitis, which is a caveat of this specific SARS-CoV-2 mouse model[27].

## Discussion

Altogether, these findings indicate that therapeutic inhibition of TBK1/IKKε as late as 3 days post-infection is superior to that of direct STING inhibition for protection against SARS-CoV-2-driven lung hyper-

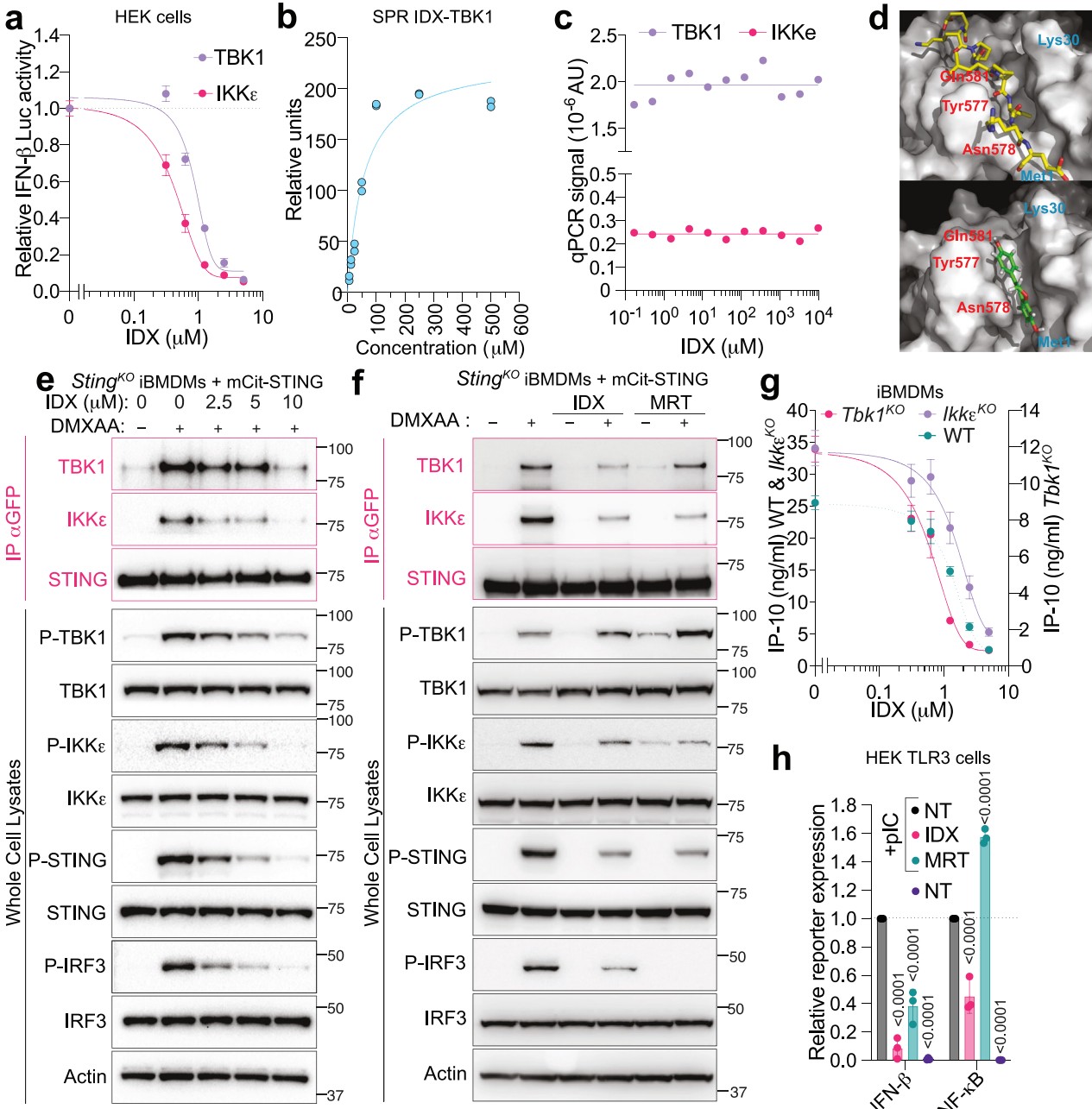

**Fig. 2 | Idronoxil inhibits formation of TBK1 signalling complexes. a** Interferon [IFN]-β luciferase expression in HEK293T cells expressing the indicated constructs after O/N incubation (*n* = 3 independent experiments). **b** Surface plasmon resonance [SPR] analysis of idronoxil (IDX) binding to recombinant human TBK1 (representative of *n* = 3 independent experiments). **c** qPCR amplification signal of kinase concentration after KinomeScan™ assays, for the indicated IDX concentrations (averaged from *n* = 2 technical replicates). IKKe is IKKε. **d** In silico modelling of IDX binding (bottom−green) and STING C-terminal tail (top−yellow, PDB: 6nt9)[19] to TBK1 dimer (the blue and red residues refer to the two different TBK1 units in the structure). Immunoblot of *Sting*^KO immortalised bone marrow-derived macrophages [iBMDMs] expressing mCitrine-STING, after 1 h stimulation with 50 µg/ml DMXAA (STING) and increasing doses of IDX (**e**), or 5 µM IDX or 100 nM MRT67307 [MRT] (**f**). **e** 1 representative blot of three independent experiments shown. **f** One

representative blot of two independent experiments shown. Pink: pull-down of mCitrine-STING with anti-GFP antibody. Black: whole cell lysates. **e, f** Phosphorylated proteins are indicated with [P]. Molecular weight markers are shown in kDa. **g** IP-10 protein levels in wild-type (IC$_{50}$ = 1.6 µM), *Tbk1*^KO (IC$_{50}$ = 0.75 µM), *Ikbke*^KO [*Ikkε*^KO](IC$_{50}$ = 2 µM) and wild-type [WT] iBMDM cells after O/N stimulation with 50 µg/ml DMXAA (STING) and increasing doses of IDX (*n* = 3 independent experiments). **h** HEK-TLR3 cells expressing NF-κB and IFN-β luciferase reporters were treated 6 h with 0.5 µg/ml poly(I:C) [pIC] (TLR3) in the presence of 5 µM IDX or 400 nM MRT (*n* = 3 independent experiments). **a, g, h** Data are mean ± s.e.m. **a**−**c, g** Non-linear regression analyses are shown. **h** Two-way ANOVA with uncorrected Fisher's LSD (with single pooled variance) multiple comparisons are shown. Exact *p* values are shown for all comparisons. Source data and detailed statistical analyses are provided as a Source Data file.

inflammation. The discrepancy between the protective effect of IDX and MRT versus H151 against hyper-inflammation in the SARS-CoV-2 mouse model demonstrates that while TBK1/IKKε are key mediators of the immunopathology, the STING pathway is likely not the dominant upstream pathway driving the damaging response. This is aligned with

the additional role for TBK1/IKKε in SARS-CoV-2-dependent MAVS inflammation[22]. However, it contrasts with the independent finding that H151 was strongly protective against lung inflammation in the same model, which could be explained by its earlier administration (from day 2) in that study[24].

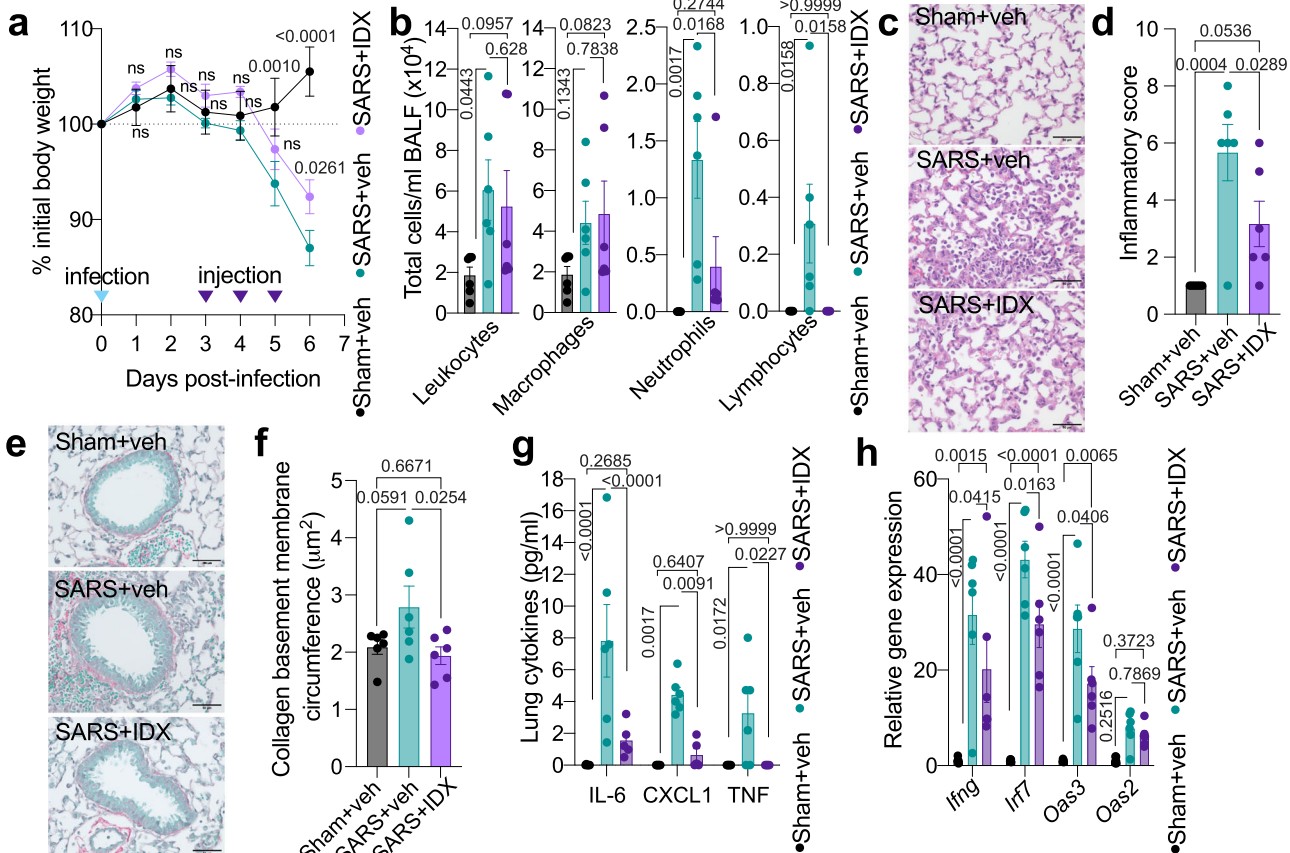

**Fig. 3 | Idronoxil limits SARS-CoV-2-driven hyper-inflammation. a** Daily weights of K18-hACE2 mice after intranasal infection (day 0) with $10^3$ PFU SARS-CoV-2 ($n = 6$ animals were examined per group in one experiment). Idronoxil [IDX]/vehicle injections were performed on days 3, 4, and 5, with mice culled on day 6 post-infection. Veh is vehicle, SARS is SARS-CoV-2 infected, and Sham is non-infected. **b** Bronchoalveolar lavage fluid [BALF] cell subsets were enumerated following Modified Giemsa staining to distinguish cell populations ($n = 6$ animals were examined per group in one experiment). **c** Histology of fixed lung lobes with Modified Giemsa staining (1 representative image of 6 per group) and (**d**) associated inflammatory scores ($n = 6$ animals were examined per group in one experiment). **e** Collagen deposition around the primary airways determined by Sirius Red/Fast Green staining (1 representative image of 6 per group) and (**f**) associated measurement of the collagen deposition around the primary airway

basement circumference ($n = 6$ animals were examined per group in one experiment). **c**, **e** Scale bars of 50 μm are indicated. **g** Cytokine quantification in lung homogenates ($n = 6$ animals were examined per group in one experiment, however one outlier in the IDX + SARS group was removed in these analyses after ROUT outlier analysis). **h** mRNA transcript levels in lung homogenates. Expression of indicated genes is shown relative to *Hprt* ($n = 6$ animals were examined per group in one experiment, however qPCR amplification failed for one animal in the Sham + vehicle group). **a**, **b**, **d**, **f**–**h** Data are mean ± s.e.m. **a**, **g**, **h** Two-way or (**b**, **d**, **f**) one-way ANOVA with uncorrected Fisher's LSD (with single pooled variance) multiple comparisons are shown. Exact *p* values are shown for all comparisons except for (**a**) where "ns" is non-significant ($p > 0.05$). Source data and detailed statistical analyses are provided as a Source Data file.

While IDX and MRT are very different molecules and display different modes of action on TBK1/IKKε, our observation that they have near indistinguishable protective activities against lung inflammation in this model establishes the potential of TBK1/IKKε as therapeutic targets in SARS-CoV-2-driven lung hyperinflammation. Our findings with SARS-CoV-2 are consistent with a previous report that selective *Tbk1*-deficiency in myeloid cells results in decreased severity of Influenza A virus infection associated with the reduced expression and production of pro-inflammatory cytokines in the BALF and lungs of infected mice[2]. Our results showing that both IDX and MRT significantly decreased TLR3-, STING-, and RIG-I-driven inflammation in primary human bronchial pulmonary cells further indicate they will likely have an anti-inflammatory function in the human lung.

Critically, late administration of IDX and MRT did not impact the viral load measured in the lungs of SARS-CoV-2-infected mice, while concurrently dampening critical inflammatory factors associated with severe COVID-19 in patients (e.g. IL-6 and IFN-γ) and decreasing airway neutrophil infiltration, which are correlated with disease severity[28–32]. The animal model employed in our study is known to result in disproportionately high numbers of brain infections and encephalitis,

which is a feature rarely observed in humans[33]. We would therefore argue that the increased viral load observed in the brains of the MRT/IDX-treated-mice is predominantly an artifact of this specific animal model and is unlikely to be physiologically relevant to the human disease. Nonetheless, further studies using aerosolised SARS-CoV-2 or alternative mouse models, which do not have brain involvement, along with strategies to promote more targeted delivery of TBK1/IKKε inhibitors to the lung, should be conducted in the future[33].

Overall, we reveal here that IDX modulates TBK1/IKKε function to restrict pro-inflammatory signalling downstream of STING, TRIF, and MAVS through its unique inhibitory effect on S172 phosphorylation[1,8,9]. While IDX may have additional anti-inflammatory activities that contribute to its protective role against SARS-CoV-2 lung inflammation, our findings that its therapeutic benefit aligns with that of MRT confirms it predominantly relies on TBK1/IKKε inhibition. To our knowledge, IDX is the first clinic-ready inhibitor capable of targeting both the IRF3 and NF-κB pro-inflammatory arms of the TBK1/IKKε axis. KINOMEscan™ analysis of the effect of 10 μM IDX on 403 kinases indicated that it was broadly a poor kinase inhibitor, only binding its top two targets (PIM2 and DRAK1) in the micromolar range

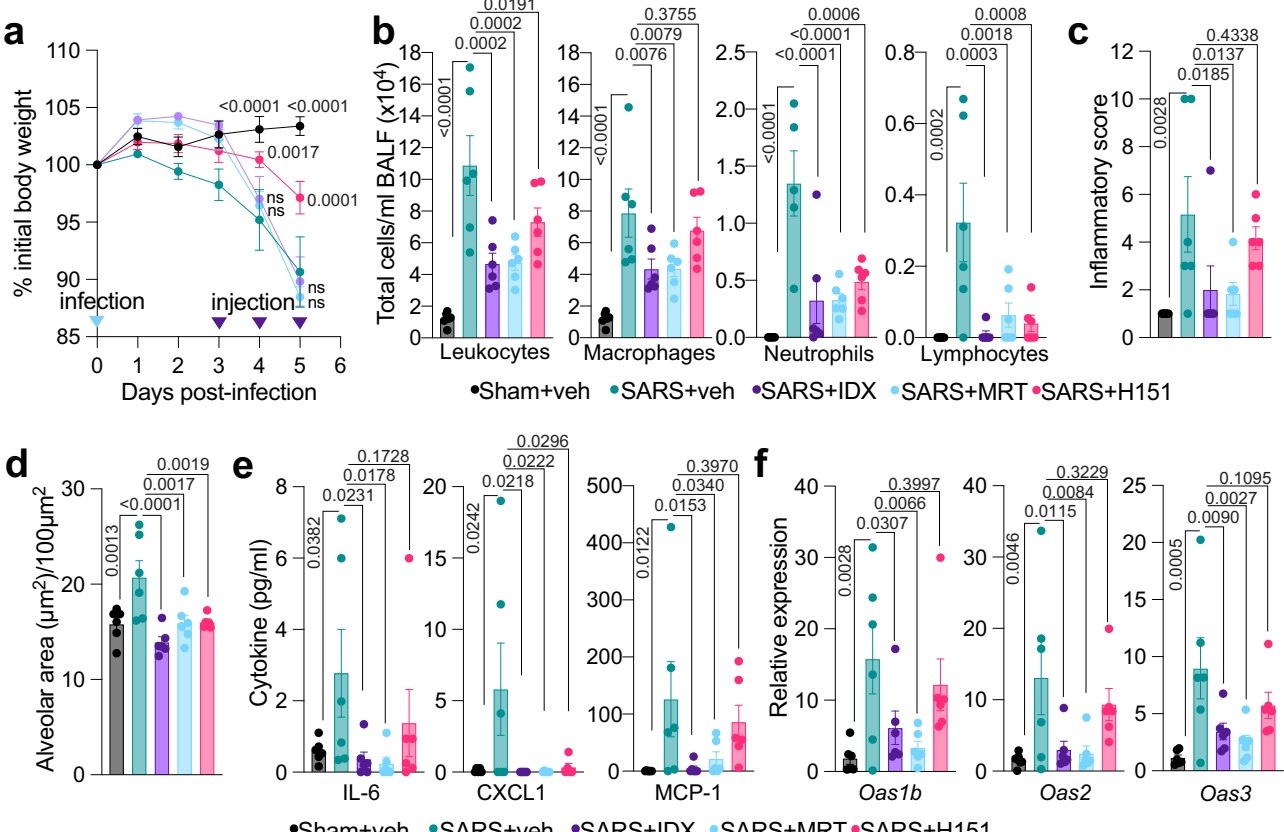

**Fig. 4 | TBK1 inhibition limits SARS-CoV-2-driven hyper-inflammation. a** Daily weights of K18-hACE2 mice after intranasal infection (day 0) with 10³ PFU SARS-CoV-2. Idronoxil [IDX]/MRT67307 [MRT]/H151/vehicle injections were performed on days 3, 4, and 5, with mice culled on day 5 post-infection ($n = 6$ animals were examined per group in one experiment). Veh is vehicle, SARS is SARS-CoV-2 infected, and Sham is non-infected. **b** Bronchoalveolar lavage fluid [BALF] cell subsets were enumerated following Modified Giemsa staining to distinguish cell populations ($n = 6$ animals were examined per group in one experiment, however one outlier in the SARS + Vehicle group was removed in the Neutrophils analyses). **c** Histology of fixed lung lobes with associated inflammatory scores ($n = 6$ animals were examined per group in one experiment). **d** Alveolar thickness in the primary airways determined by histology ($n = 6$ animals were examined per group in one

experiment). **e** Cytokine quantification in lung homogenates ($n = 6$ animals were examined per group in one experiment, however one outlier in each but the SARS + Vehicle group was removed in the CXCL1 analyses after ROUT outlier analysis). **f** mRNA transcript levels in lung homogenates. Expression of indicated genes is shown relative to *Hprt* ($n = 6$ animals were examined per group in one experiment). **a**–**f** Data are mean ± s.e.m. **a** Two-way or (**b**–**f**) one-way ANOVA with uncorrected Fisher's LSD (with single pooled variance) multiple comparisons are shown. Exact $p$ values are shown for all comparisons except for (**a**) where the comparisons for days 0, 1, 2 and 3 are provided in Source Data File, and where "ns" is non-significant ($p > 0.05$). Source data and detailed statistical analyses are provided as a Source Data file.

(Supplementary Table 2). In stark contrast, TBK1 kinase inhibitors (e.g., MRT or BX795), have nanomolar affinities for >20 other kinases within a 10-fold window of their TBK1 binding affinity[34], suggestive of more frequent off-target effects. Together with its excellent safety profile in humans (established in over 600 cancer patients[35]), our preliminary Phase I clinical study of rectal delivery of IDX supported good tolerability in 38 patients hospitalised due to moderate COVID-19 disease[36]. Therefore, given IDX's unique mechanism-of-action in restricting harmful TBK1/IKKε signalling, we anticipate that our findings will be the starting point in the therapeutic development of IDX, or its derivatives, to limit the harmful TBK1/IKKε signalling seen in viral-mediated hyper-inflammation, including COVID-19 immunopathology.

# Methods
## Ethics
All the animal experiments complied with relevant local ethical regulations (Institutional Animal Care and Use Committee of Shanghai Chempartner, the Sydney Local Health District Institutional Biosafety and Animal Ethics Committees, the Monash University Animal Ethics committee regulations and the Australian National University animal ethics). The Monash Health and Monash Medical Centre Human

Research Ethics Committee approved the studies conducted on human tissues.

## Cell culture and reagents
HEK-cGAS^low cells overexpressing murine cGAS and constitutively producing cGAMP, HEK-STING cells stably express murine STING protein fused to an N-terminal mCherry-tag, and L929 cells expressing an IFN-stimulated response element [ISRE]-Luciferase (also referred to as LL171 cells) have all been previously described[37]. p125HEK cells stably expressing an IFN-β-Luc reporter were previously described[38]. HEK 293 cells stably expressing human TLR3 (HEK-TLR3, #hkb-htlr3) were from Invivogen. WT, *Sting^KO*, *Tbk1^KO*, *Ikbkε^KO*, and *Tbk1^KO/Ikbkε^KO* iBMDMs were generated previously[9]. Human BJ-5ta hTERT foreskin fibroblasts (referred to as hTERT fibroblasts herein) were obtained from ATCC (#RL4001). HEK293T cells stably expressing EGFP[39] were used in TBK1/IKKε overexpression experiments. HEK-cGAS^low, HEK-STING, p125HEK, L929, WT/*Sting^KO/Tbk1^KO/Ikbkε^KO*/ *Tbk1^KO-Ikbkε^KO* iBMDMs, and HEK293T cells were cultured in Dulbecco's modified Eagle's medium plus L-glutamine supplemented with 1× antibiotic/antimycotic (Thermo Fisher Scientific) and 10% heat-inactivated fetal bovine serum (FBS) (referred to as complete DMEM hereafter). HEK-

TLR3 were maintained in complete DMEM supplemented with 30 µg/ml Blasticidin (Invivogen). WT THP-1 cells[40] were grown in RPMI 1640 plus L-glutamine (Life Technologies) supplemented with 1x antibiotic/antimycotic and 10% heat-inactivated FBS (complete RPMI). hTERT fibroblasts were grown in complete DMEM supplemented with 10 µg/ml Hygromycin B. Vero cells (Creative Biolabs, #CAR-STC-ZP9) used in SARS-CoV-2 infections were grown in M199 medium (Gibco) supplemented with 5% FBS (Gibco) and 1X p/s (Gibco). All cell lines were cultured at $37\,°C + 5\%\ CO_2$. Cell lines were passaged 2–3 times/week and tested for *Mycoplasma* contamination on a routine basis by PCR or using Mycostrip (Invivogen).

Human primary bronchial epithelial cells (PBEC) were obtained and cultured as described previously[41]. The Monash Health and Monash Medical Centre Human Research Ethics Committee approved the studies; consent was obtained from all subjects, and studies were conducted in accordance with the approved guidelines. Briefly, PBECs were obtained from bronchial brushings during routine bronchoscopy and cultured under submerged conditions on collagen-coated flasks (MP Biomedicals) in supplemented bronchial epithelial growth medium (BEGM; Lonza). When PBEC reached 80% confluency they were treated with IDX/MRT and further stimulated as indicated.

*Trex1^{Q98X}* C57BL/6NCrl mice (used under Australian National University animal ethics, reference A2018/38) have a single base-pair mutation in *Trex1* leading to a premature stop codon (Q98X). Homozygous mutant mice aberrantly accumulate cytoplasmic DNA, resulting in basal engagement of the cGAS-STING pathway (Ellyard J.I. and Vinuesa C.G., manuscript in preparation), similar to that reported in TREX1-deficient mice[16]. Primary BMDMs from three 10-week old TREX1-mutant male mice were isolated and differentiated for 6 days in complete DMEM supplemented with L929-conditioned medium[42].

Resveratrol (#R5010), quercetin (#337951), apigenin (#A3145), and daidzein (#D7802) were all purchased from Sigma and were resuspended at 50 mM in DMSO (stored at −20 °C). Idronoxil and derivatives (see Supplementary Methods) were synthesised by Noxopharm Ltd and resuspended at 40 mM in DMSO (stored at −20 °C). diABZI compound #3 (GSK3 herein) (Selleckchem #S8796) and ADUS100 (MedChemExpress #HY-12885) were used as human STING agonists, and DMXAA (Cayman chemical #14617) as a mouse STING agonist. The human STING inhibitor H151 (Cayman chemical #25857) was used where indicated. The TBK1 inhibitor, MRT67307, was from MedChemExpress (#HY-13018). The TLR4 agonist, LPS (#LPS-EB) and TLR3/RIG-I/MDA5 agonist high molecular weight (HMW) polyI:C (#tlrl-pic), and the tri-phosphate RNA selective RIG-I agonist (#3p-hpRNA) were from Invivogen. The cGAS ligand ISD70 was resuspended/annealed as per[40]. Remdesivir (GS-5734) was from Selleckchem. ISD70 was transfected at 2.5 µg/ml at a ratio of 1 µg:1 µl with Lipofectamine 2000 in Opti-MEM (Thermo Fisher Scientific). Poly(I:C) was transfected at 1 µg/ml at a ratio of 1 µg:2.5 µl with Lipofectamine 2000 in Opti-MEM. 3p-hpRNA was transfected at 50 ng/ml at a ratio of 40 ng:1 µl with Lipofectamine 2000 in Opti-MEM.

For Fig. 1b, c, e–h; Ext. Data Figs. 1b–m and 2e–h; Ext Data Fig. 2e–g; IDX and other compounds were added 1 h prior to cell stimulation (with STING agonist or other ligands).

## Plasmacytoid dendritic cells studies

C57BL/6J mice, at 6–8 weeks of age were housed at Monash University Animal Facility and all procedures were carried out according to Monash University Animal Ethics committee regulations. Dendritic cells (DCs) from spleens were isolated as per[43]. DCs were resuspended in RPMI-1640 Glutamax (Thermo Fisher Scientific) supplemented with 10% FBS (In vitro Technologies), 100 µM 2-Mercaptoethanol (Thermo Fisher Scientific) and 0.01% Penicillin/Streptomycin (Thermo Fisher Scientific). DCs were first incubated at $3.0 × 10^6$ cells/ml in media alone or with IDX at concentrations of 1.25–2.5 µM, for 1 h at $37\,°C + 10\%\ CO_2$ and then stimulated with media alone, 10 µg/ml R848 (InvivoGen) or 0.5 µM diABZI (GSK3) at a final concentration of $1.5 × 10^6$ cells/ml at $37\,°C + 10\%\ CO_2$ for 18–20 h. Supernatants were analysed for plasmacytoid DC-produced IFN-α using a flow cytometric bead-based assay LEGENDplex™ kit (BioLegend) according to manufacturer's instructions.

## In vitro antiviral assays

These experiments were conducted by Creative Biolabs. The virus used was the SARS-CoV2 isolate England/02/2020 (BEI Resources NR52359). Vero cells were seeded in complete M199 medium at 8000 cells/100 µl/well, and incubated until the next day at 37 °C, 10% $CO_2$. The virus stock was diluted from a stock of $1.7 · 10^7$ IU/ml to reach a multiplicity of infection (MOI) of 0.002. The compounds were assayed in a 3-fold serial dilution (from 50 µM for IDX, and 20 µM for Remdesivir) in biological triplicate. The virus was added to the cells for 1 h, prior to treatment with IDX or Remdesivir, and further incubation for 24 h at 37 °C, 10% $CO_2$. After 24 h, the plates were washed with PBS, fixed for 30 min with 4% formaldehyde, washed again with PBS, and stored in PBS at 4 °C until staining. Any residual formaldehyde was quenched with 50 mM ammonium chloride, after which cells were permeabilized (0.1% Triton X-100) and stained with an antibody recognising SARS-CoV2 spike protein (GeneTex GTX632604). The primary antibody was detected with an Alexa-488 conjugated secondary antibody (Life Technologies, A11001), and nuclei were stained with Hoechst. Images were acquired on a CellInsight CX5 high content platform (Thermo Scientific) using a 4× objective, and percentage infection was calculated using CellInsight CX5 software (infected cells/total cells × 100). Cytotoxicity was assessed in parallel using the MTT assay (Sigma, M5655).

## Luciferase assays

For overexpression of human TBK1, or human IKKε (in pcDNA3.1, a kind gift from A. Mansell, Hudson Institute), 400 ng of respective vector was co-transfected with 200 ng of reporter (IFN-β-Luc reporter plasmid (a kind gift from K. Fitzgerald, University of Massachusetts) or pNF-κB-Luc4 reporter (Clontech), in $7.5 × 10^5$ HEK293T-EGFP cells with lipofectamine 2000 in a 6-well plate. Cells were washed 3 h after transfection, plated into 12 wells of a 96-well plate, and treated with the indicated amounts of IDX overnight. A similar protocol was used for HEK-STING cells co-transfected with IFN-β-Luc and the murine cGAS-GFP construct[44] (a kind gift from V. Hornung, University of Munich). The next day, cells were lysed in 40 µl (for a 96-well plate) of 1× Glo Lysis buffer (Promega). For LL171 or p125HEK cells treated with DMXAA or 3p-hpRNA, respectively, the cells were lysed in 40 µl of 1× Glo Lysis buffer. Lysates (15 µl) were then subjected to a firefly luciferase assay using 40 µl of Luciferase Assay Reagent (Promega). Luminescence was quantified with a Fluostar OPTIMA (BMG LABTECH) luminometer with OPTIMA-Control v2.2R2 software. MARS Data analysis software 3.01R2 (BMG Labtech) was used for data analyses.

## Cytokine analysis

Production of murine IP-10, TNF and IL-6 in iBMDM cell supernatants was quantified using the CXCL10/IP-10/CRG-2 Duo Set ELISA (R&D systems; #Dy466) or BD OptEIA ELISA kits (BD Biosciences #558534 and #555240), respectively. Similarly, human IP-10, IL-6 or IFN-β levels were measured in supernatants from hTERT or THP-1 cells using the IP-10 (BD Biosciences, #555157 and #555220) and IFN-β (PBL assay science, #41415-1) ELISA kits, respectively. Tetramethylbenzidine substrate (Thermo Fisher Scientific) was used for quantification of the cytokines on a Fluostar OPTIMA (BMG LABTECH) plate-reader with OPTIMA-Control v2.2R2 software. MARS Data analysis software 3.01R2 (BMG Labtech) was used for data analyses. All ELISAs were performed according to the manufacturers' instructions.

## Immunoblotting

For Fig. 1e, f and Supplementary Figs. 1m and 2f: 1–1.5 × 10⁶ iBMDMs were lysed with 150 µl RIPA buffer supplemented with PMSF, 1× phosSTOP and 1× cOmplete™ protease inhibitors (Roche). Lysates were cleared by spinning at 17,000 × *g* through Pierce centrifuge columns (Thermo Fisher Scientific #89868) for 1 min before addition of 4× SDS-PAGE sample loading buffer and 10 min denaturation at 95 °C. For Supplementary Fig. 1g, 1 × 10⁵ hTERT fibroblasts were lysed with 120 µl 1× NuPAGE LDS buffer (supplemented with 10% reducing agent, 1× phosSTOP and 1× cOmplete™ protease inhibitors). Lysates were heated at 70 °C for 10 min, centrifuged briefly, and 20–25 µl loaded on Bolt™ 8% Bis-Tris Mini Protein Gels[9]. The following antibodies from Cell Signalling were used: STING (#13647S, 1:1000), p-STING (#72971 and #19781, both at 1:500), TBK1 (#3013, 1:1000), p-TBK1 (#5483, 1:500), IKKε (#3416 and #2690, 1:1000), p-IKKε (#8766, 1:500), IRF3 (#4302, 1:1000), p-IRF3 (#4947, 1:500), p65 (#4764, 1:1000), and p-p65 (#3033, 1:500). Anti-β ACTIN (Abcam #Ab49900, 1:10,000) was used as loading control. Peroxidase-AffiniPure Goat Anti-Rabbit (Jackson ImmunoResearch Labs #111-035-003, 1:10,000) was used as secondary antibody for Chemiluminescent detection (as described in ref. 9). Goat anti-Mouse IgG (H + L) Highly Cross-Adsorbed Secondary Antibody, HRP Thermo Fisher Scientific Cat# A16078, used at 1:1000 dilution. Acquisition of images and analyses was conducted with Image Lab 6.1 (Bio-Rad). Uncropped and unprocessed scans of the representative blots shown in the Figures are available in the Source Data file or at the end of the Supplementary Information file.

## Isolation of STING–TBK1 complexes by immunoprecipitation

Immunoprecipitation experiments were performed similarly to those previously described[45]. Approximately 15 × 10⁶ *Sting*^KO iBMDMs expressing mCitrine-mouse STING were lysed on ice for 30 min with 1 ml of 1× NP-40 buffer (1% Nonidet-P40, 20 mM Tris-HCl pH 7.4, 150 mM NaCl, 1 mM EGTA, 10% glycerol, 10 mM NaPPi, 5 mM NaF and 1 mM Na₃VO₄) supplemented with 1 mM PMSF and cOmplete™ protease inhibitors (Roche Biochemicals). Whole cell lysates were subsequently clarified by centrifugation at 13,000 × *g* for 10 min at 4 °C. Following preparation of samples for immunoblot, 1 µg of anti-GFP antibody (Thermo Scientific; clone E36, A-11120) was added to the remaining whole cell lysate. Samples were then incubated at 4 °C for up to 2 h on a rotator before 50 µl of Dynabeads Protein G (Thermo Scientific; 10004D) were added. Samples were then incubated once more at 4 °C for up to 2 h on a rotator before beads were extensively washed with lysis buffer using a DynaMag-2 magnet (Thermo Scientific; 12321D). Proteins were eluted from beads by addition of 35 µl of 2× reducing SDS-PAGE sample loading buffer (2.5% SDS, 25% glycerol, 125 mM Tris-HCl pH 6.8, 0.01% bromophenol blue, 100 mM dithiothreitol) and heating at 95 °C for 10 min. Samples were then subjected to immunoblotting with acquisition of images and analyses with Image Lab 6.1 (Bio-Rad)[9,46]. Uncropped and unprocessed scans of the representative blots shown in the Figures are available in the Source Data file.

## Cloning, expression, and purification of TBK1 protein

The full-length human TBK1 sequence was cloned into pFastBac-His-TEV (N-terminal hexahistidine tag and TEV protease cleavage site) vector by ChemPartner. Following 48 h expression at 27 °C in 0.7 l of Sf9 insect cells, the cells were pelleted and lysed (20 mM Tris, pH 8.0, 300 mM NaCl, 20 mM Imidazole, 0.1 mM benzonase Cocktail). His-TEV-TBK1 was purified using Ni-FF column and subsequently digested by TEV protease to remove the affinity tag. The tag removed TBK1 was further purified by size exclusion chromatography using a S200 Superdex column in elution buffer (20 mM HEPES, pH 7.5, 300 mM NaCl, 0.5 mM TCEP). Purified TBK1 protein was concentrated, dispensed as aliquots at 1 mg/ml, and frozen at −80 °C. Protein identity and purity were assessed by SDS-PAGE and SEC-HPLC and mass spectrometry.

## RNA and RT-qPCR analyses from cell-based experiments

Total RNA was purified from *Trex1*^Q98X primary BMDMs or human PBECs using the ISOLATE II RNA Mini Kits (Bioline). Random hexamer cDNA was synthesised from isolated RNA using the High-Capacity cDNA Archive kits (Thermo Fisher Scientific) according to the manufacturer's instructions. RT-qPCR was carried out with the Power SYBR Green Master Mix (Thermo Fisher Scientific) on a QuantStudio 6 RT-PCR system (Thermo Fisher Scientific) and analysed with the Quant-Studio Real-Time PCR Software v1.7.2. Each PCR was performed in technical duplicate and human *18S* was used as the reference gene (Fig. 1h). Each amplicon was gel-purified and used to generate a standard curve for the quantification of gene expression. Melting curves were used in each run to confirm specificity of amplification. Primers used were the following: Human *IFNB1*: hIFNB-FWD GCTTGGATTCCT ACAAAGAAGCA; hIFNB-REV: ATAGATGGTCAATGCGGCGTC; Human *IFIT1*: IFIT1-FWD TCACCAGATAGGGCTTTGCT; hIFIT1-REV CACCTCAA ATGTGGGCTTTT; Human *RSAD2*: hRSAD2-RT-FWD TGGTGAGGTT CTGCAAAGTAG; hRSAD2-RT-REV GTCACAGGAGATAGCGAGAATG; Human *18S*: h18S-FWD CGGCTACCACATCCAAGGAA; h18S-REV GCTGGAATTACCGCGGCT.

## RNA sequencing

Libraries were generated using an in-house multiplex RNA-seq method (MHTP, Medical Genomics Facility, as per[47]). Libraries were prepared using 10 ng of total RNA. An 8 bp sample index and 10 bp unique molecular identifier (UMI) were added during initial poly(A) priming and pooled samples were amplified using a template-switching oligonucleotide. Illumina P5 and P7 sequences were added by PCR and Nextera transposase, respectively. The library was designed so that the forward read (R1) utilised a custom primer (5′ GCC TGT CCG CGG AAG CAG TGG TAT CAA CGC AGA GTA C 3′) to sequence directly into the index and then the 10 bp UMI. The reverse read (R2) used the standard R2 primer to sequence the cDNA in the sense direction for transcript identification. Paired-end sequencing (R1 19 bp; R2 72 bp) was performed on a NextSeq 2000 (Illumina) P2 run. Base calling was performed using Dragen BCLConvert (v3.7.4).

## RNA-seq analysis

RNA-seq analysis was performed in R (v4.1.0)[48]. The scPipe package (v1.14.0)[49] was employed to process and de-multiplex the data. A combined multiplexed FASTQ file was created from the R1 and R2 FASTQ files by trimming the sample index and UMI sequences and storing them in the read header using the sc_trim_barcode function (with bs2 = 0, bl2 = 8, us = 8, ul = 10). Read alignment was performed on the combined multiplexed FASTQ file using the Rsubread package (v2.6.1)[50]. An index was built using the Ensembl *Mus musculus* GRCm39 primary assembly genome file and alignment was performed with default settings. Aligned reads were mapped to exons using the sc_exon_mapping function with the Ensembl *Mus musculus* GRCm39 v104 GFF3 genome annotation file. The resulting BAM file was de-multiplexed and reads mapping to exons were associated with each individual sample using the sc_demultiplex function, taking the UMI into account, and an overall count for each gene for each sample was generated using the sc_gene_counting function (with UMI_cor = 1). Additional gene annotation was obtained using the biomaRt package (v2.48.2)[51] and a DGEList object was created with the counts and gene annotation using the edgeR package (v3.34.0)[52]. A design matrix was constructed incorporating the treatment group and donor mouse (BMDMs from *n* = 3 mice were used). Lowly expressed genes were removed using the filterByExpr function and normalisation factors were calculated using the TMM method[53]. Counts were transformed

using the voom method[54] and a linear model was fit using the edgeR voomlmFit function.

Differential gene expression analyses were performed using the limma (v3.48.3) package[55]. IDX treatment was compared to control using the contrasts.fit function and moderated *t*-statistics were calculated using the treat function[56] with a 1.2-fold cut-off. Differentially expressed genes were determined using a false discovery rate (FDR) adjusted *p* value < 0.05. Gene set tests were performed using the cameraPR function[57], with the *t*-statistics recalculated without a fold-change threshold using the eBayes function[58], and the Broad Institute Molecular Signatures Database Hallmark gene set collection[59], which was obtained using the msigdbr package (v7.4.1)[60].

To obtain a list of top IFN-stimulated genes (ISGs), all mouse type I IFN datasets were taken from the Interferome v2.0 database (http://interferome.its.monash.edu.au/interferome/home.jspx)[61]. For each dataset, significantly up-regulated genes (log₂ fold change > 1; unadjusted *p* < 0.05) were ranked by the magnitude of their log₂ fold changes. These ranks were then combined across datasets by geometric mean using the TopKLists package (v1.0.7)[62] to produce an overall ranked list of type I ISGs. The heat map displaying the top 20 ISGs was made using the pheatmap package (v1.0.12)[63]. Log₂ counts per million (CPM) expression values were calculated using the edgeR cpm function. Relative log₂ CPM expression values were calculated by subtracting the average value across all samples calculated using the aveLogCPM function, with the heat map scale truncated to ±2.

The original multiplexed R1 and R2 FASTQ files were de-multiplexed using cutadapt v2.10[64] (with error rate 0 and action none) and deposited along with the UMI counts in the NCBI Gene Expression Omnibus (GEO) with accession GSE193353.

## KINOMEscan™ TBK1 and IKKε kinase assays

The KINOMEscan™ kinase assays[65] were performed by Eurofins DiscoverX Corporation. For each kinase, the assay relies on the ability of the tested compound to compete against an immobilised ligand for binding to the DNA-tagged kinase. RT-qPCR is then used to measure the quantity of kinase molecules bound to the immobilised ligand, which is decreased if the tested compound competed with the kinase-ligand interaction. An 11-point 3-fold serial dilution of IDX compound was prepared in 100% DMSO at 100× final test concentration and subsequently diluted to 1× in the assay.

## Molecular docking studies

Docking of IDX to the dimer of TBK1 was performed using the software quCBit (MedChemSoft Solutions, version 2019) and the crystal structure of the TBK1 dimer (PDB: 6nt9)[19].

## IDX tissue distribution

All the procedures related to animal handling, care and the treatment in this study were performed according to the guidelines approved by the Institutional Animal Care and Use Committee (IACUC) of Shanghai Chempartner following the guidance of the Association for Assessment and Accreditation of Laboratory Animal Care. Mouse lung and plasma samples were collected from PC-3 tumour-bearing ~9–11 week old male BALB/c nude mice that had been treated with 20 mg/kg or 50 mg/kg of IDX twice per day for 21 days, at 2 h post-final dose (*n* = 3) or 12 h post-final dose (*n* = 3). IDX was administered via intraperitoneal (i.p.) injection in a formulation of 5% DMSO, 10% Solutol HS15 and 85% saline at 2 mg/ml for the 20 mg/kg dosing group and 5 mg/ml for the 50 mg/kg dosing group. Lung samples were homogenised with 10 volumes (v/w) of PBS. A 30 μl aliquot of plasma or homogenised lung sample was added to 200 μl IS (glipizide, 40 ng/ml) and mixed by vortexing for 1 min before centrifugation at 3200 × *g* for 10 min. Supernatants (100 μl) were transferred to new plates. Standard working solutions were prepared in 1:1 (v/canACN:H₂O) before mixing 3 μl of working solution with 57 μl of blank matrix to obtain a calibration range of 2–3000 ng/ml. Analysis was performed using LC-MS/MS with separation performed on a Waters ACQUITY UPLC HSS T3 (2.1 × 50 mm, 1.8 μm) column with 0.70 ml/min flow rate, column temperature of 50 °C, and gradient mobile phase (mobile phase A: H₂O, 5 mm NH₄OAc; mobile phase B: MeOH, 5 mm NH₄OAc). Detection was performed using SCIEX LC-MS/MS-37 (Triple Quad 6500) using ESI in negative mode with MRM detection.

## SARS-CoV-2 mouse infection

SARS-CoV-2 mouse experiments shown in Figs. 3 and 4 were approved by the Sydney Local Health District Institutional Biosafety and Animal Ethics Committees and were performed under BSL3 conditions. SARS-CoV-2 Wuhan isolate (VIC01/2020) was propagated in VeroE6 cells to generate high titre stocks[25,66]. Locally bred female hemizygous K18-hACE2 mice[67] (B6.Cg-Tg(K18-ACE2)2Prlmn/J) at ~8 weeks of age were transported to the Centenary Institute PC3 facility for SARS-CoV-2 infection. Mice were anaesthetised using isoflurane and were intranasally inoculated with 10³ PFU SARS-CoV-2 (VIC01/2020) in 30 μl volumes[25,66]. Mice were housed using the IsoCage N biocontainment system (Tecniplast) and were weighed and monitored daily[25,66]. IDX (20 mg/kg/day), MRT (15 mg/kg/day)[68], H-151 (750 nmol/day)[24] or vehicle were administered via i.p. injection on days 3, 4 and 5 in a formulation of 5.33% DMSO, 4% Solutol HS15 (#HY-Y1893-50G, from MedChemExpress) and 90.67% saline, in 6 mice per group. At day 5 or 6 post-inoculation (as specified), mice were euthanised with an overdose of sodium pentobarbitone (Virbac). BALF and lung tissues were collected and processed as per[25,66]. Viral titres in lung homogenates from infected mice were quantified using VeroE6 plaque assays[25,66]. Viral titres in the brain from infected mice were similarly quantified using VeroE6 plaque assays collected from half the brain tissue dissected down the midline and homogenised as lung tissue for viral titre quantification. BALF cell pellets were deposited onto glass microscope slides using a Cytospin centrifuge (10 g, 7 min), left to dry overnight and then stained using Quick Dip Stain Kits (Modified Giemsa Stain) according to the manufacturer's instructions (POCD Scientific). BALF cell pellet populations were enumerated via microscopy of cells on haematoxylin and eosin stained slides based on morphological characteristics. Formalin-fixed lung tissues were paraffin embedded, sectioned, and stained with either Quick Dip Stain Kit (POCD Scientific) or Sirius Red/Fast Green Stain (Sigma-Aldrich) for inflammatory score and collagen deposition, respectively, as per the manufacturer's instructions. Inflammatory score and collagen deposition were quantified as per[69]. Alveolar area was quantified in ImageJ (v1.53) using histological microscopy images of the lung parenchyma and was calculated as a measurement of the total alveolar area in each field of view normalised to an area of 100 μm². Cytokine quantification in the BALF supernatant and lung homogenates were determined using Cytometric Bead Arrays (Becton Dickinson) as per manufacturer's instructions[66]. To determine the relative expression of IFN and antiviral genes, total RNA was extracted from lung tissue homogenised in TRIzol reagent (Sigma-Aldrich). cDNA synthesis was performed using M-MLV Reverse Transcriptase as per the manufacturer's instructions (Thermo Fisher). qRT-PCR was performed using iTaq™ Universal SYBR® Green Supermix (Bio-Rad) and a CFX384 Touch Real-Time PCR Detection System (Bio-Rad) and analysed with CFX Maestro Software V2.1 (Bio-Rad). Primer sequences used were: *Ifngr1* (Fwd: 5'- ACAG CTCTCCGTCCTCGTAT-3', Rv: 5'- CACTCCGGTTATGCTCCACA), *Ifnb1* (Fwd: 5'- AACTCCACCAGCAGACAGTG-3', Rv: 5'- GGTACCTTTGCAC CCTCCAG-3'), *Ifnl3* (Fwd: 5'- CTTCAGGCCACAGCAGAGCCCAAG, Rv: 5'- ACACACTTGAGGTCCCGGAGGA-3'), *Ifng* (Fwd: 5'- GAGGAACTGGC AAAAGGATGG-3', Rv: 5'- TGCTGATGGCCTGATTGTCTT-3'), *Ifnar2* (Fwd: 5'- GACCCCGCAATAAAATCTCCCT-3', Rv: 5'- GCAGCTCAGTG GTGTGCATTTA-3'), *Irf7* (Fwd: 5'- CCAGCCACGGAAAATAGGGA-3', Rv: 5'- CCCGGCATCACTAGAAAGCA-3'), *Oas1b* (Fwd: 5'- TCTGCTTTA TGGGGCTTCGG-3', Rv: 5'- TCGACTCCCATACTCCCAGG-3'), *Oas2*

(Fwd: 5′-GTGACATGGTGGGAGTGTTCA-3′, Rv: 5′- CCGGGGGTCTGC ATTACCTA-3′), *Oas3* (Fwd: 5′- AGGCTACCGTGTACGCATCT-3′, Rv: 5′- T TCACACAGCGGCCTTTACC -3′), and *Mx2* (Fwd: 5′- TTCACCAGGC TCCGAAAAGA-3′, Rv: 5′- AGCTGGTTCTTCCAGGGTTT-3′). Fold change expression of genes of interest were compared against the expression of a reference gene; *Hprt* (Fwd: 5′- AGGCCAGACTTTGTTGGATTTGA A-3′, Rv: 5′- CAACTTGCGCTCATCTTAGGCTTT-3′), and analysed using the ΔΔCt method.

## Surface plasmon resonance

Surface plasmon resonance (SPR) was performed using a Biacore T200 instrument (Cytiva) at 25 °C in a buffer containing 10 mM HEPES (pH 7.4), 150 mM NaCl, 0.5 mM TCEP and 0.005% Tween-20 (HBS). Ten μg of TBK-1 at a concentration of 9 mg/ml was diluted into 10 mM sodium acetate pH 4.0 and immobilised on a carboxymethylated dextran CM5 chip (Cytiva) by amine coupling, with a similarly activated and deactivated adjacent flow cell with no coupled protein serving as a control. IDX dissolved in DMSO to make up a 10 μM stock solution, which was then serially diluted in HBS and injected at a flow rate of 30 μl/min for 120 s. Following subtraction of data from the control flow cell and the result from injection of HBS with 0 mM IDX, the steady-state $K_D$ values were derived by the steady-state affinity option of the Biacore evaluation software Version 3.2 (Cytiva) and Prism (GraphPad Prism software version 9.31).

## Statistical analyses

Statistical analyses were carried out using Prism 10.0.1 (GraphPad Software Inc.). One-way and two-way analyses of variance (ANOVA) with uncorrected Fisher's LSD were used when comparing groups of conditions, unless otherwise stated, while unpaired two-tailed *t*-tests were used when comparing pairs of conditions. Cell-based experiments were conducted at least two independent times in biological replicates (unless otherwise stated). Detailed statistical analyses used in each Figure are provided as a Source Data file.

## Reporting summary

Further information on research design is available in the Nature Portfolio Reporting Summary linked to this article.

## Data availability

The data that support this study are available from the corresponding author upon request. The RNA sequencing data generated in this study have been deposited NCBI Gene Expression Omnibus (GEO) with accession GSE193353. The interferome v2 database is available at http://interferome.its.monash.edu.au/interferome/home.jspx. The raw data including uncropped and unprocessed scans of the representative blots shown in the Figures and each line graphs and bar charts generated in this study are provided in the Supplementary Information/Source Data file. Source data are provided with this paper.

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

## Acknowledgements

We thank V. Hornung for the cGAS-GFP expressing vector, HEK-cGAS^low, and HEK-STING cells; J. Rehwinkel for the p125HEK cells; A. Mansell for the TBK1 and IKKε vectors; K. Fitzgerald for the pLuc-IFN-β reporter; M. Pelegrin for the LL171 cells; T. Wilson and the MHTP Genomics platform for RNA sequencing; Matthew L. Dennis for help with protein work; M. Speir for helping with the preparation of the manuscript; Jubilant Biosys Ltd. and Shanghai ChemPartner Co., Ltd. for performing the compound synthesis and in vivo pharmacokinetics studies, respectively. Eurofins Scientific for the KINOMEscan analyses. This work was supported by the

Australian National Health and Medical Research Council Project Grant (1124485 to M.P.G.; 2011467 to M.D.J. and N.G.H.; 1175134 to P.M.H.) and Program Grant (1113577 to B.T.K.); the Australian Research Council (140100594 Future Fellowship to M.P.G.); the Kenyon Foundation (to M.D.J.); the Victorian Cancer Agency (mid-career fellowship to J.P.V.); the Rainbow Foundation, NSW RNA Production Network and UTS (to P.M.H.); a Monash Silver Jubilee Postgraduate Research scholarship (M.S.J.S.) and Monash Graduate Excellence scholarship (to K.B.); a Monash University FMNHS Senior Postdoctoral Fellowship (to D.D.N.); the Victorian Government's Operational Infrastructure Support Program and COVID-19 Treatments Medical Research Fund; and Noxopharm Limited.

## Author contributions

Conceptualisation: T.R.U., M.D.J., D.S.W., O.F.L., P.M.H., D.D.N., and M.P.G.; Investigation: T.R.U., M.D.J., K.R.B., R.L.A., L.J.G., D.S.W., J.Z., J.R., J.P.V., S.S., W.S.N.J., R.R., S.M., D.H.N., N.G.H., R.V., V.R.A., J.H.K., E.S.P., B.T., A.S.A., W.A.D., J.I.E., W.W., D.D.N., and M.P.G.; Resources: M.O.K., N.K., B.T.K., C.G.V., G.E.K., P.M.H., O.F.L., D.D.N., and M.P.G.; Data curation: L.J.G. and M.P.G.; Writing—original draft: T.R.U., M.D.J., D.D.N., and M.P.G.; Writing—review and editing: M.D.J., K.R.B., L.J.G., D.S.W., J.Z., J.I.E., W.W., N.K., B.T.K., C.G.V., P.M.H., O.F.L., D.D.N., and M.P.G.; Supervision: P.M.H., D.D.N., and M.P.G.; Project administration: T.R.U., M.D.J., D.S.W., O.F.L., P.M.H., D.D.N., and M.P.G.; Funding acquisition: J.P.V., M.O.K., G.E.K., O.F.L., P.M.H., D.D.N., and M.P.G.

## Competing interests

Financial competing interests: O.F.L. and D.S.W. are employees of Noxopharm. G.E.K. owns equity in Noxopharm. M.P.G., J.Z., J.I.E., V.R.A., N.K. and P.M.H. receive funding from Noxopharm Ltd. to study the activity of IDX in inflammation. M.P.G. and P.M.H. received consulting fees from Noxopharm Ltd. Noxopharm Ltd. was involved in the conceptualisation, design, data collection, analysis and helped the preparation of the manuscript. M.P.G., J.Z., V.R.A., N.K., J.I.E. and P.M.H. do not personally own shares and/or equity in Noxopharm Ltd. G.E.K., O.F.L. and M.P.G. are named inventors of a patent (assignee: Noxopharm Ltd) relating to the use of IDX to treat inflammation associated with infection (WO/2021/195698, including patent US20210299085 granted). The remaining authors declare no competing interests.

## Additional information

[1]Centre for Innate Immunity and Infectious Diseases, Hudson Institute of Medical Research, Clayton, VIC, Australia. [2]Department of Molecular and Translational Science, Monash University, Clayton, VIC, Australia. [3]Centre for Inflammation, Centenary Institute and University of Technology Sydney, Faculty of Science, School of Life Sciences, Sydney, NSW, Australia. [4]Department of Biochemistry and Molecular Biology, Monash Biomedicine Discovery Institute, Monash University, Clayton, VIC, Australia. [5]St. Vincent's Institute of Medical Research, Fitzroy, VIC, Australia. [6]Department of Medicine, The University of Melbourne, Melbourne, VIC, Australia. [7]Noxopharm Limited, Chatswood, NSW, Australia. [8]School of Chemistry, UNSW Sydney, Kensington, NSW, Australia. [9]MedChemSoft Solutions, Ferntree Gully, VIC, Australia. [10]Monash Lung and Sleep, Monash Medical Centre, Clayton, VIC, Australia. [11]Department of Clinical Laboratory Sciences, College of Applied Medical Sciences, Taif University, Turabah, Saudi Arabia. [12]Department of Immunology and Infectious Diseases, John Curtin School of Medical Research, Australian National University, Canberra, ACT, Australia. [13]Centre for Personalised Immunology, John Curtin School of Medical Research, Australian National University, Canberra, ACT, Australia. [14]Centre for Cancer Research, Hudson Institute of Medical Research, Clayton, VIC, Australia. [15]Faculty of Health and Medical Sciences, University of Adelaide, Adelaide, SA, Australia. [16]Francis Crick Institute, London, UK. [17]These authors contributed equally: Tomalika R. Ullah, Matt D. Johansen. [18]These authors jointly supervised this work: Philip M. Hansbro, Dominic De Nardo, Michael P. Gantier. ✉e-mail: Michael.gantier@hudson.org.au

