## [Peer Review File · Nature Communications]

Pharmacological inhibition of TBK1/IKK ϵ blunts immunopathology in a murine model of SARS-CoV-2 infectionEditorial Note: This manuscript has been previously reviewed at another journal that is not operating a transparent peer review scheme. This document only contains reviewer comments and rebuttal letters for versions considered at *Nature Communications*.

REVIEWER COMMENTS

Reviewer #2 (Remarks to the Author):

The authors have answered to my remaining questions.

Reviewer #3 (Remarks to the Author):

In principle, I am now convinced that the authors have identified an IKKe /TBK1 inhibitor. However, it remains a major problem that the whole paper is structured with focus on STING signaling, and then the authors end up testing a model, where they basically demonstrate that the pathology blocked by the novel inhibitor is only to a limited extent STING driven. Therefore, I suggest that Fig 1 and 2 should be restructured and re-described, with broader focus on 2-3 innate immune pathways, this leading to the conclusion that IDX targets several pathways. This is what is actually found, and the most fair way to describe the data.

Associated with this (and also related to my previous point on the limited basic scientific novelty of the data), I think the authors should be very bold in their conclusions (abstract, and discussion), that the discrepancy between the effect of IDX and H151 in the COVID-19 mouse models, demonstrate that while TBK1/IKKe are major signaling proteins driving pathology in the disease, the STING pathway is likely not a major driver of this response.

Additional comment:

This reviewer was additionally asked to comment in the absence of reviewer 1 on their prior criticisms and the response of the authors to these concerns.

The authors have in my opinion done a good attempt to improve the work, and it has been improved! However, I still think we are at the lower limit of what should be published in *Nature Communications* (both from the standpoint of novelty, fundamental advancement in immunology, mechanistic insight into the MOA of the compound). I would recommend rejection, but acknowledge it is a difficult one, since there is also strong stuff in the manuscript.

The authors have tried to address the points raised by reviewer #1, but there are still several outstanding points. So, yes advancement has been made, but it can be debated whether it is sufficient

Response to Reviewers – Manuscript NI-A33791B/NCOMMS-23-00142-T

We sincerely thank the Editor and Reviewers for their assessment of our work. We have thoroughly addressed the comments in the point-by-point response below. The Reviewer comments are presented in *bold italics*, while our responses are in blue. Note that the new edits of our manuscript have been highlighted in blue, while preserving the yellow changes from the previous reviews.

Reviewer #2:

The authors have answered to my remaining questions.

We thank Reviewer 2 for their positive assessment of our revision. We have added the following paragraph in relation to one of their previous concerns:

“We note that the affinity of the interaction between IDX and TBK1 was relatively low in view of its potency in the micromolar range in cells. While this discrepancy could relate to conformational changes of TBK1 induced through interaction with other protein binding partners in cells, and which are lacking in our in vitro assays, these data confirmed that IDX can bind TBK1.”

Reviewer #3:

In principle, I am now convinced that the authors have identified an IKKe /TBK1 inhibitor. However, it remains a major problem that the whole paper is structured with focus on STING signaling, and then the authors end up testing a model, where they basically demonstrate that the pathology blocked by the novel inhibitor is only to a limited extent STING driven. Therefore, I suggest that Fig 1 and 2 should be restructured and re-described, with broader focus on 2-3 innate immune pathways, this leading to the conclusion that IDX targets several pathways. This is what is actually found, and the the most fair way to describe the data.

We have now entirely re-structured and re-written the description of previous Figures 1, 2 and 3. Since the previous Figures 1 and 2 were both focussed on the modulation of STING signalling in mouse and human cells, respectively, we merged the essential results of these two figures within Figure 1 and Supplementary Figure 1. Notably, we now introduce the capacity of IDX to inhibit TLR3, RIG-I (with a new Figure [Fig.1g] demonstrating the dose-dependent effect of IDX on a selective RIG-I agonist) and TLR4 in Figure 1 and Supplementary Figure 1, leading to the conclusion that IDX appears to target the common downstream effectors, TBK1/IKKe. Indeed, we further demonstrate the direct effect of IDX on TBK1 and IKKe in Figure 2 and Supplementary Figure 2, and its non-classical inhibitory activity, which distinguishes it from traditional kinase inhibitors such as MRT67307.

Associated with this (and also related to my previous point on the limited basic scientific novelty of the data), I think the authors should be very bold in their conclusions (abstract, and discussion), that the discrepancy between the effect of IDX and H151 in the COVID-19 mouse models, demonstrate that while TBK1/IKKe are major signaling proteins driving pathology in the disease, the STING pathway is likely not a major driver of this response.

We thank the reviewer for this suggestion and have added two paragraphs to directly make this point:

Abstract: *“Our additional findings demonstrate that engagement of STING is not the major driver of these inflammatory responses...”*

Discussion: *“The discrepancy between the protective effect of IDX and MRT versus H151 against hyper-inflammation in the SARS-CoV-2 mouse model demonstrates that while TBK1/IKKe are key mediators of the immunopathology, the STING pathway is likely not the dominant upstream pathway driving the damaging response.”*

Additional comment:

The authors have tried to address the points raised by reviewer #1, but there are still several outstanding points. So, yes advancement has been made, but it can be debated whether it is

sufficient

Referee 1 had two major criticisms during our previous review:

A) there is still no convincing evidence to suggest that IDX inhibits SARS-CoV-2-induced inflammatory response through suppressing STING/TBK1 SARS-CoV-2-induced inflammation.

As pointed out by Reviewer 3, our results suggest that STING is not a predominant driver of the COVID19 immunopathology since TBK1 inhibition by IDX and MRT is more protective than that of STING inhibition. Consequently, we have now restructured the manuscript to remove the focus on IDX modulating STING in COVID19 immunopathology. The new version describes that our findings rather support the activity of IDX on TBK1/IKKe, which suppresses SARS-CoV-2-induced inflammation.

B) The authors attempted to validate this experimentally but failed to optimise the pull-down conditions successfully in order to obtain meaningful results (using overexpression of tagged constructs).

Although we were unable to optimise pull-down of TRIF/TBK1 and MAVS/TBK1 we did successfully show that pull-down of STING/TBK1 was impacted by IDX, and that this was different from the effect of MRT. Along with the SPR data showing direct binding of IDX to TBK1, we also demonstrated that IDX significantly blunted signalling responses driven by TLR3/4/TRIF and RIG-I/MAVS. Critically, we showed that phosphorylation of TBK1 and IKKe was dampened by IDX upon TLR4 and STING stimulation, confirming that the effect of IDX on S172 phosphorylation was consistently dampened by IDX independent of the upstream pathway engaged.

REVIEWERS' COMMENTS

Reviewer #3 (Remarks to the Author):

The authors have now addressed my points in a satisfactory manner.

Response to Reviewers – Manuscript NCOMMS-23-00142-A

We sincerely thank the Editor and Reviewers for their assessment of our work. The Reviewer comments are presented in ***bold italics***, while our responses are in blue. Note that the new edits of our manuscript to address the editorial comments have been highlighted in Green, while preserving the yellow and blue changes from the previous reviews.

Reviewer #3:

The authors have now addressed my points in a satisfactory manner.

We thank Reviewer 3 for their positive assessment of our revision.